# Molecular Basis Underlying the Therapeutic Potential of Vitamin D for the Treatment of Depression and Anxiety

**DOI:** 10.3390/ijms23137077

**Published:** 2022-06-25

**Authors:** Bruna R. Kouba, Anderson Camargo, Joana Gil-Mohapel, Ana Lúcia S. Rodrigues

**Affiliations:** 1Center of Biological Sciences, Department of Biochemistry, Federal University of Santa Catarina, Florianópolis 88040-900, SC, Brazil; rkba.bruna@gmail.com (B.R.K.); camargo.andersonc@gmail.com (A.C.); 2Island Medical Program, Faculty of Medicine, University of British Columbia, Victoria, BC V8P 5C2, Canada; 3Division of Medical Sciences, University of Victoria, Victoria, BC V8P 5C2, Canada

**Keywords:** anti-inflammatory effect, anxiety, depression, neuromodulator, pro-neurogenic effect, vitamin D

## Abstract

Major depressive disorder and anxiety disorders are common and disabling conditions that affect millions of people worldwide. Despite being different disorders, symptoms of depression and anxiety frequently overlap in individuals, making them difficult to diagnose and treat adequately. Therefore, compounds capable of exerting beneficial effects against both disorders are of special interest. Noteworthily, vitamin D deficiency has been associated with an increased risk of developing depression and anxiety, and individuals with these psychiatric conditions have low serum levels of this vitamin. Indeed, in the last few years, vitamin D has gained attention for its many functions that go beyond its effects on calcium–phosphorus metabolism. Particularly, antioxidant, anti-inflammatory, pro-neurogenic, and neuromodulatory properties seem to contribute to its antidepressant and anxiolytic effects. Therefore, in this review, we highlight the main mechanisms that may underlie the potential antidepressant and anxiolytic effects of vitamin D. In addition, we discuss preclinical and clinical studies that support the therapeutic potential of this vitamin for the management of these disorders.

## 1. Introduction

Major depressive disorder (MDD) and anxiety disorders, which in this review will be referred to as depression and anxiety, are devastating and highly prevalent clinical entities that constitute one of the leading causes of disability worldwide [1,2]. These disorders often occur concomitantly and their symptoms frequently overlap in individuals, and patients with these two comorbidities often present with a higher severity and duration of symptoms [3]. Therefore, the diagnosis and treatment of these disorders remain a challenge in the clinical setting [1,4]. Despite being different disorders, the etiology of depression and anxiety involves similar factors, such as genetic predispositions, environmental aspects, and several biological mechanisms [1,2,5]. Among the main biological mechanisms implicated in the pathophysiology of these disorders, compelling evidence has pointed to neuroinflammation as a key factor in the onset and progression of these disorders [6]. Notably, other biological mechanisms that have been implicated in depression and anxiety, such as gut dysbiosis, impaired neurogenesis, and monoaminergic dysfunction, may be triggered by a neuroinflammatory process, opening new perspectives for studying molecular targets and neuroprotective agents against these mood disturbances [4,7,8,9]. In this regard, in recent years, vitamin D has gained prominence due to its antioxidant, anti-inflammatory, pro-neurogenic, and neuromodulatory properties that appear to be fundamental to its antidepressant and anxiolytic effects [10,11,12,13]. Given this background, in this review, we highlight the main mechanisms that may underlie the potential antidepressant and anxiolytic effects of vitamin D. In addition, we discuss preclinical and clinical studies that support the therapeutic potential of this vitamin for the management of these mood disorders. 

## 2. Literature Data Searching

This review presents a mechanistic overview of the clinical and preclinical research regarding the effects of vitamin D on depression and anxiety. The search included original manuscripts and contemporary reviews published in English, assessed by specific search terms in the title or abstract of the manuscripts available through PubMed. The search terms used were “vitamin D”, “calcitriol”, “cholecalciferol”, “calcidiol”, “anxiety”, “depression”, “major depressive disorder”, “neuroinflammation”, “inflammation” “neurogenesis”, and “monoaminergic system”. We performed a specific screening of the clinical and preclinical studies that investigated the role of vitamin D in anxiety and depression. To review the molecular basis underlying the therapeutic potential of vitamin D for the treatment of depression and anxiety, we selected preclinical and clinical studies published over a 28-year period (1994 to 2022).

## 3. Neuroinflammation as a Key Pathophysiological Mechanism Related to Mood Disorders

Neuroinflammation is a complex process that comprises a defense mechanism in the central nervous system (CNS), protecting and restoring the structure and function of the brain against infection and injury, through the modulation of neurogenesis, axonal regeneration, and remyelination of neural cells [14]. However, chronic and exacerbated inflammatory responses can produce harmful effects in the brain. These inflammatory processes may involve inflammation-related signaling molecules, microbiota, as well as immune and brain cells [7,15].

Microglia, nervous-system-specific immune cells, are of special interest to neuroinflammatory responses. Under normal conditions, microglia assume a phenotype defined by a ramified morphology and highly motile processes for constant monitoring of the brain parenchyma (M2 phenotype). After an insult, promoted by pathogen-associated molecular patterns and/or damage-associated molecular patterns that interact with pattern recognition receptors (PRRs), such as Toll-like receptors (TLRs), microglia retract their processes and adopt an ameboid form (M1 phenotype) [16,17,18,19]. In addition to morphological changes, the binding of these molecular patterns to these receptors induces the priming of NLRP3 [nucleotide-binding oligomerization domain (NOD)-, leucine-rich repeats (LRR)- and pyrin domain-containing protein 3] and pro-interleukin (IL)-1β expression via nuclear factor kappa B (NF-kB) and myeloid differentiation primary response 88 (Myd88) pathways [17,20,21]. Noteworthily, the expression of NLRP3 may be inhibited under certain conditions, particularly upon activation of nuclear factor erythroid 2-related factor 2 (Nrf2), the main regulator of the antioxidant response [22,23]. Subsequently, different stimuli capable of promoting mitochondrial dysfunction, calcium and potassium ion flux, reactive oxygen species (ROS) production, and lysosomal damage activate the NLRP3 inflammasome, promoting the autoproteolytic activation of pro-caspase-1 [20]. Caspase-1 can subsequently cleave pro-interleukin-1β and pro-interleukin-18 into their active forms, interleukin-1β (IL-1β) and interleukin-18 (IL-18), respectively [17,24]. In addition, activation of NLRP3 can lead to gasdermin D-mediated formation of membrane pores and subsequently pyroptosis [17]. 

Mediators released by activated microglia induce astrocyte polarization [25]. This polarization contributes to the impairment of signaling pathways that play a crucial role in neuronal survival and synaptic plasticity, such as the brain-derived neurotrophic factor (BDNF)/tropomyosin-related kinase B (TrkB) signaling pathway [26]. Chronic activation of astrocytes also results in increased levels of chemokines, such as chemokine ligand 2, which interact with peripheral immune cell receptors to induce infiltration of macrophages and monocytes from the circulation into the CNS and resulting in increased blood–brain barrier permeability [27,28].

Interestingly, neuroinflammatory-process-derived pro-inflammatory cytokines, such as interferon-gamma (IFN-γ) and tumor necrosis factor-alpha (TNF-α), cause an increase in the activity of the enzyme indoleamine 2,3-dioxygenase in astrocytes, microglia, and inflammatory cells. The activation of this enzyme increases the formation of quinolinic acid, an *N*-methyl-D-aspartate (NMDA) receptor agonist, through the kynurenine pathway, contributing to glutamatergic system disturbance and compromising the synthesis of serotonin by depleting tryptophan [8,29,30,31]. Within this scenario, it is worth noting that more than 90% of the body’s serotonin is produced in the gut, particularly by enterochromaffin cells, and alterations in the intestinal microbiota triggered by inflammatory processes have been shown to directly compromise the synthesis of this monoamine [32,33]. Indeed, it is important to note that changes in gut microbiota have been reported to impair the efficacy of antidepressants, such as fluoxetine [34]. 

In summary, neuroinflammation results from intrinsic communication among peripheral immune cells, gut microbiota, and immune cells present in the brain, and these interactions culminate in glutamatergic dysregulation, as well as reduced synaptic plasticity and monoamine synthesis. Together, these factors contribute to the atrophy and neuronal loss seen in patients with chronic depression and anxiety disorders [35]. In this regard, compounds with anti-inflammatory, pro-neurogenic, and neuromodulatory properties, such as vitamin D, are of special interest for the treatment of these mood disorders. 

## 4. Metabolism and Biological Functions of Vitamin D

Vitamin D is a steroid hormone that can be obtained from the diet in two forms: (i) vitamin D_2_ (ergocalciferol) found in yeast, mushrooms, as well as plants, and (ii) vitamin D_3_ (cholecalciferol) found in foods of animal origin, such as cod liver oil and oily fish [36]. However, the main source of vitamin D is cholecalciferol synthesized endogenously through solar ultraviolet B (UVB) radiation at wavelengths of 290–315 nm [36,37]. Through this process, UVB photons penetrate into the dermis and are absorbed by 7-dehydrocholesterol, which is then converted into previtamin D_3_ [37]_._ Previtamin D_3_ subsequently isomerizes into vitamin D_3_, which is then transported in the circulation by vitamin D-binding protein (DBP) [37]. In the liver, vitamin D_3_ is hydroxylated at C-25 by 25-hydroxylase, mainly by cytochrome P450 Family 2 Subfamily R Member 1 (CYP2R1), to produce 25-hydroxyvitamin D_3_ (25(OH)D_3_) [38,39]. Following this, 25(OH)D_3_ can be stored or can undergo a second hydroxylation in the kidneys by 25-hydroxyvitamin D-1α-hydroxylase (Cytochrome P450 Family 27 Subfamily B Member 1; CYP27B1), forming its biologically active form, 1α,25-dihydroxyvitamin D (1,25(OH)_2_D_3_), also known as calcitriol [40]. It is estimated that humans can synthesize quantities equivalent to the oral intake of 20,000 IU of this vitamin [38]. 

Vitamin D is widely known to play critical roles in calcium–phosphorus metabolism, acting in the maintenance of bone homeostasis [38]. In addition, vitamin D also exerts pleiotropic functions that have gained increased attention in the last few years [41]. These biological actions of calcitriol are mediated by vitamin D receptors (VDRs) [42]. The interaction between calcitriol and VDR allows other transcription factors, such as the retinoid X receptor (RXR), to interact with this complex, forming a heterodimer [42]. This heterodimer can then bind to vitamin D-responsive elements (VDREs), and by recruiting complexes of either co-activators or co-repressors, it can modulate gene expression [42,43]. It is estimated that 200 to 2000 genes are modulated directly or indirectly by vitamin D [42]. Due to the numerous biological actions mediated by vitamin D, the measurement of its serum concentration becomes indispensable. In this regard, the Endocrine Society’s Clinical Practice Guideline defines vitamin D deficiency as values <20 ng/mL; insufficiency between 21–29 ng/mL; and sufficiency between 30–100 ng/mL [36,40]. However, reference values for vitamin D remain a matter of debate [44]. 

### 4.1. Antioxidant and Anti-Inflammatory Properties of Vitamin D

Neurons and glial cells express VDR and vitamin D-metabolizing enzymes in various regions, such as the prefrontal cortex and hippocampus, suggesting a possible role for vitamin D in depression and anxiety disorders [45]. In line with this, several studies have suggested an inverse association between serum vitamin D levels and symptoms related to depression and anxiety [46,47,48]. In addition to glial cells, VDRs and CYP27B1 are also expressed in several other immune cells, such as CD4^+^ and CD8^+^ T cells, B cells, macrophages, neutrophils, and dendritic cells [49]. Notably, it has been shown that calcitriol is capable of modulating innate and adaptive immune responses [49]. Furthermore, the positive modulation of antioxidant enzymes by calcitriol is known to contribute to redox homeostasis and, consequently, to the attenuation of the neuroinflammatory process [50,51]. For example, calcitriol has been shown to inhibit the synthesis of interleukin interleukin-6 (IL-6), TNF-α, and nitric oxide (NO) by activated microglia [12]. Likewise, it has been shown to decrease the expression of IL-1β, TNF-α, and inducible nitric oxide synthase (iNOS), the enzyme responsible for the synthesis of NO under inflammatory conditions, in the prefrontal cortex of rats [52]. 

In addition to IL-6 and TNF-α, vitamin D_3_ has also been shown to reduce the expression of interleukin-12 (IL-12) and increase the expression of the anti-inflammatory cytokine interleukin-10 (IL-10) [53]. Furthermore, vitamin D was shown to up-regulate the expression of other anti-inflammatory cytokines, including interleukin-14 (IL-4), and transforming growth factor-beta (TGF-β) [54]. Calcitriol is also capable of up-regulating TLR10 while down-regulating TLR2 and TLR4, reinforcing its role in attenuating microglial activation [55,56]. 

In this context, it has been reported that vitamin D mitigates the activation of the NLRP3 inflammasome [57,58]. Although a study by Camargo et al. (2020) did not find changes in NLRP3 expression in mice treated for 7 days with cholecalciferol, this treatment reduced the immunocontent of other inflammasome-related proteins, including ASC (apoptosis-associated speck-like protein containing a caspase recruitment domain), TXNIP (thioredoxin interacting protein), and caspase-1 in the hippocampus of mice. However, Xin et al. (2019) showed that calcitriol attenuated NLRP3 expression by inhibiting the NF-kB pathway in human bronchial epithelial cells. Likewise, treatment with a VDR agonist (paricalcitol) was able to attenuate the expression of NLRP3, gasdermin D, caspase-1, and IL-1β, proteins related to the NF-kB-mediated pyroptosis process in cisplatin-induced acute kidney injury models [59]. One of the possible mechanisms by which vitamin D may reduce the activation of NF-kB is by up-regulating the NF-kB inhibitor IkBα (nuclear factor of kappa light polypeptide gene enhancer in B-cells inhibitor, alpha) [60]. In addition, it has been reported that VDR can bind NLRP3, inhibiting the activation of this complex [61,62]. 

Calcitriol is also able of suppressing NLRP1 inflammasome activation by up-regulating the nuclear factor erythroid 2-related factor 2 (Nrf2) signaling pathway [50]. Indeed, the ability of calcitriol to activate Nrf2 signaling seems to be directly associated with its neuroprotective properties [63]. Once activated, Nrf2 promotes the transcription of several antioxidant enzymes [64]. In line with this, studies have shown that vitamin D is able to induce the expression of heme oxygenase-1 (HO-1), catalase, superoxide dismutase (SOD), and glutamate-cysteine ligase (the rate-limiting enzyme in the synthesis of glutathione, an endogenous antioxidant), thereby inhibiting ROS accumulation in the brain [51,65,66,67]. 

### 4.2. Pro-Neurogenic and Neuromodulatory Properties of Vitamin D

Vitamin D deficiency during pregnancy has been demonstrated to directly affect brain development in the offspring [68,69,70]. Vitamin D-deficient offspring exhibit a thinner cortex, reduced expression of genes and proteins related to cytoskeleton maintenance, synaptic plasticity, neurotransmission, cell proliferation, and growth, as well as decreased levels of nerve growth factor (NGF) [70,71,72,73]. Although no alterations in BDNF and glial cell line-derived neurotrophic factor (GDNF) levels were found in vitamin D-deficient offspring, different lines of evidence have shown that this vitamin is effective in inducing the synthesis of these neurotrophins [74,75]. In addition, vitamin D also participates in the synthesis of the neurotrophin NT-3, a key regulator of survival, growth, and differentiation of neurons [76,77]. 

Indeed, neurotrophins bind to receptor tyrosine kinases (Trk), which, in turn, contribute to promoting the survival and growth of neurons as well as synaptogenesis [78]. In this regard, a study by Atif et al. showed that a combined treatment with progesterone and vitamin D was effective in activating the BDNF/TrkB signaling pathway [79]. Although this study did not explore the mechanisms underlying this, it is known that activated TrkB can phosphorylate and activate the mammalian target of rapamycin (mTOR), resulting in the translation of synaptic proteins, including postsynaptic density protein-95 kDa (PSD-95), glutamate α-amino-3-hydroxy-5-methyl-4-isoxazolepropionic acid (AMPA) receptor subunit 1 (GluA1), and synapsin [80,81]. Noteworthily, synapsins can increase neurotransmitter release in the synaptic cleft [82]. Interestingly, vitamin D supplementation has been shown to promote an increase in serum serotonin in individuals suffering from depression [83] as well as serum dopamine levels in children with attention-deficit/hyperactivity disorder (ADHD) [84]. Within this scenario, it has been shown that vitamin D can directly increase the biosynthesis of monoamines by augmenting the expression of the enzymes tyrosine hydroxylase and tryptophan hydroxylase, which mediate the synthesis of dopamine/noradrenaline and serotonin, respectively, particularly in the prefrontal cortex [85]. In addition, VDR overexpression has been shown to increase dopaminergic neuron differentiation, and consequently, dopamine production [86]. Moreover, Morello et al. (2018) showed that calcitriol can cause neuronal proliferation in primary cultures of murine neural progenitor cells and improve neurogenesis in transgenic Alzheimer’s disease (AD)-like mice [5XFAD model, which expresses human amyloid precursor protein (APP) and presenilin 1 (PSEN1) transgenes with a total of five AD-linked mutations: the Swedish (K670N/M671L), Florida (I716V), and London (V717I) mutations in APP, and the M146L and L286V mutations in PSEN1). Finally, calcitriol was also able to inhibit serotonin reuptake transport and the expression of the monoamine oxidase-A (MAO-A) gene, suggesting that this vitamin and antidepressants may share common mechanisms of action [87]. 

### 4.3. Vitamin D: Modulation of Gut Microbiota

The immunological and neuromodulatory role of vitamin D occurs, in part, through its interaction with the gut microbiota [88]. Indeed, serum vitamin D concentration may contribute to changes in the composition of gut microbiota. Furthermore, vitamin D deficiency, VDR, or CYP27B1 depletion have been shown to result in an increase in the Bacteroidetes and Proteobacteria phyla, triggering epithelial barrier dysfunction and intestinal inflammation [89,90]. Interestingly, VDR has been shown to negatively regulate bacteria-induced intestinal NF-kB activation, attenuating inflammatory responses. Therefore, VDR is an important contributor to intestinal homeostasis and bacterial infection protection [91]. The main neuromodulatory, anti-inflammatory, and antioxidant properties of vitamin D are illustrated in Figure 1. 

## 5. Preclinical Studies: Effects of Vitamin D in Models of Depression and Anxiety

In recent years, several preclinical studies have been conducted to investigate the possible antidepressant and anxiolytic effects of vitamin D (Table 1).

Of note, chronic administration of cholecalciferol [5 mg/kg for 14 days; subcutaneous (s.c.) administration] elicited an antidepressant-like effect in the forced swim test in ovariectomized Wistar rats [92]. In addition, cholecalciferol supplementation (5 mg/kg for 14 days; s.c.) was capable of attenuating anxiety-like behaviors in the elevated plus-maze and the light–dark box tests in ovariectomized Wistar rats [10,93]. Based on these findings, further studies have been conducted to better understand the mechanisms underlying the anxiolytic and antidepressant effects of vitamin D in animal models [94,98]. 

Camargo et al. (2018) reported that cholecalciferol [2.5 µg/kg, orally by mouth (p.o.)], administered once a day in the last 7 days of chronic corticosterone administration (20 mg/kg, p.o., for 21 days), exerted an antidepressant-like effect in male mice subjected to the splash test and tail suspension test. Additionally, in this study, cholecalciferol treatment attenuated the increase in protein carbonyl and nitrite levels induced by corticosterone in the brain, suggesting that vitamin D_3_ has an antidepressant-like effect by, in part, modulating oxidative stress [94]. The administration of cholecalciferol (100 IU/kg, p.o.) for 7 days also abolished the depressive-like behavior in the tail suspension test induced by chronic corticosterone administration in female mice [95]. In this study, a significant decrease in ROS production in the hippocampus was observed after treatment with cholecalciferol, both in control and corticosterone-exposed mice, reinforcing the notion that the antidepressant-like effect of this vitamin may involve the modulation of oxidative stress [95]. More recently, a study by Neis et al. (2022) showed that repeated administration of cholecalciferol for 7 days (2.5 μg/kg, p.o.) abolished chronic unpredictable stress-induced depressive-like behavior in the tail suspension test, as well as a reduction in serotonin levels in the prefrontal cortex of female mice. Moreover, reinforcing the involvement of the serotonergic system in the antidepressant-like effect of cholecalciferol, in this study, the administration of the serotonin synthesis inhibitor *p*-chlorophenylalanine methyl ester was effective in abolishing the reduction in immobility time in the tail suspension test elicited by cholecalciferol [96]. 

The repeated administration of a low dose of cholecalciferol (2.5 μg/kg, p.o.) also caused an antidepressant-like effect and was effective in reducing the immunocontent of proteins that form the NLRP3 inflammasome, such as ASC [apoptosis-associated speck-like protein containing a caspase recruitment domain (CARD)], caspase-1, and thioredoxin-interacting protein (TXNIP) in the hippocampus of male mice [57]. Calcitriol treatment (100 ng/kg, p.o.) for 10 weeks in ovariectomized female Sprague–Dawley rats was also effective in producing neuroprotective effects by regulating the adenosine monophosphate (AMP)-activated protein kinase (AMPK)/NF-kB signaling pathway. Moreover, calcitriol treatment reduced the pro-inflammatory cytokines IL-1β, IL-6, and TNF-α, as well as iNOS and cyclooxygenase-2 (COX-2) levels in the hippocampus [97]. This suggests that the modulation of the NLRP3 inflammasome-driven pathway may underlie, at least in part, the antidepressant-like effect of this vitamin. 

More recently, Bakhtiari-Dovvombaygi et al. (2021) also reported that the anti-inflammatory and antioxidant effects displayed by pretreatment with vitamin D_3_ (10,000 IU/kg for 28 days) in male rats underlie the ability of this vitamin to abrogate anxiety- and depressive-like behaviors induced by chronic unpredictable mild stress (CUMS) in the elevated plus-maze and forced swimming test. Indeed, these protective effects of vitamin D were accompanied by a decrease in cortical malondialdehyde and IL-6 levels, as well as an increase in total thiol levels and enhanced SOD and catalase activity [98]. Interestingly, another study observed that following 4 weeks of CUMS, the occurrence of depressive-like behaviors was associated with an increase in 1,25(OH)_2_D and VDR expression in the hippocampus of rats, suggesting a compensatory mechanism, by which vitamin D may protect against the development of depressive-like behaviors [99]. 

In addition to attenuating depressive- and anxiety-like behaviors through its anti-inflammatory and antioxidant properties, the modulation of neurotrophic factors has also been shown to contribute to the protective properties of vitamin D [76]. Xu et al. used male C57BL/6 mice to show that calcitriol (25 μg/kg/day for 4 weeks; i.c.v.) is effective in acting as an antidepressant in a post-stroke depression model by up-regulating VDR and BDNF expression [75]. In an ovariectomized Wistar rat model of depression induced by CUMS, vitamin D_3_ (5 mg/kg for 4 weeks; s.c.) treatment was able to reverse depression-like behaviors in the sucrose preference test and the forced swimming test by increasing BDNF and NT-3/NT-4 levels in the hippocampus [76]. Although these studies showed that vitamin D supplementation results in an increase in pro-neurogenic neurotrophins, such as BDNF, Groves et al. observed that vitamin D deficiency in BALB/c mice was associated with depressive-like behaviors without compromising hippocampal neurogenesis [100]. Therefore, further studies are needed to elucidate whether neurogenesis is critical for the anxiolytic and antidepressant effects of this vitamin.

## 6. Clinical Studies: Effects of Vitamin D in Depression and Anxiety

Several studies have reported that vitamin D supplementation improves symptoms of depression and anxiety associated with various medical conditions, including type II diabetes, Crohn’s disease, ulcerative colitis, and obesity [101,102,103,104,105]. 

However, the potential therapeutic effects of vitamin D in individuals primarily diagnosed with depression or anxiety remain controversial. For example, vitamin D supplementation (1600 IU for 6 months) was shown to significantly improve anxiety symptoms, but not depressive symptoms, in patients with vitamin D deficiency [106]. Likewise, supplementation with 2800 IU of vitamin D in patients with depression did not promote a significant reduction in Hamilton D-17 scores [107]. On the other hand, supplementation with 50,000 IU of vitamin D for 2 weeks was able to improve depression severity, as assessed with the Beck Depression Inventory-II (BDI-II), although no changes in serotonin levels were detected [108]. However, in another study, cholecalciferol treatment (50,000 IU for 3 months) significantly increased serum serotonin levels, while decreasing BDI scores in women with moderate, severe, and extreme depression. Interestingly, among men, an improvement in the severity of depressive symptoms with vitamin D supplementation was only observed in those diagnosed with severe depression [83]. Beneficial effects of vitamin D (50,000 IU for 8 weeks) supplementation have also been observed in older adults (over 60 years of age) with depression [109]. However, lower doses of vitamin D (400 IU daily for 2 years) were not able to improve depressive symptoms [110]. Finally, a single dose of vitamin D (300,000 IU) was reported as an effective and safe intervention in MDD with concurrent vitamin D deficiency [111,112]. In patients with depression, the daily administration of 1500 IU vitamin D_3_ plus 20 mg fluoxetine for 8 weeks was superior to fluoxetine alone [113]. Another study reported that vitamin D supplementation (50,000 IU once/week for 3 months) in combination with standard of care improved the severity of anxiety in individuals diagnosed with Generalized Anxiety Disorder by increasing serotonin concentrations and decreasing the levels of the inflammatory biomarker neopterin [114]. Overall, although there is compelling clinical evidence pointing to the benefits of vitamin D for the management of depression and anxiety, it is important to note that divergent results have also been obtained [46]. Multiple factors may contribute to these discrepant results, including differences in the doses of vitamin D used, treatment time, serum 25-hydroxyvitamin D levels at baseline, nutritional condition at the onset of the treatment, age and sex of the individuals, as well as the presence of comorbidities that may influence the efficacy of vitamin D supplementation. Additionally, the heterogeneity observed in clinical studies may also be associated with genetic polymorphisms that may affect vitamin D efficacy [115,116,117].

## 7. Conclusions and Future Directions

In this review, we discussed the main evidence underlining the therapeutic potential of vitamin D in the management of depression and anxiety disorders. Of particular relevance, compelling evidence suggests that vitamin D possesses antioxidant, anti-inflammatory, pro-neurogenic, and neuromodulatory properties, and, thus, may act in a similar manner to classic antidepressants. In support, several preclinical studies have shown the beneficial effects of vitamin D supplementation in animal models of these mood disorders. Given this, some studies have investigated the possible synergistic or additive beneficial effects of vitamin D and current pharmacotherapy for the management of anxiety and depression. However, the clinical studies that assess the efficacy of vitamin D supplementation in the treatment of depression and anxiety are still scarce and their results are, at times, controversial. The discrepant results from clinical trials call for the need to conduct future studies, so as to establish a protocol for vitamin D supplementation that is effective in preventing or attenuating depressive and anxiety symptoms. Therefore, appropriately randomized clinical trials assessing the potential of vitamin D as co-adjuvant for the treatment of these mood disorders are warranted, so as to ascertain the true therapeutic value of this vitamin in the context of depression and anxiety. 

## Figures and Tables

**Figure 1 ijms-23-07077-f001:**
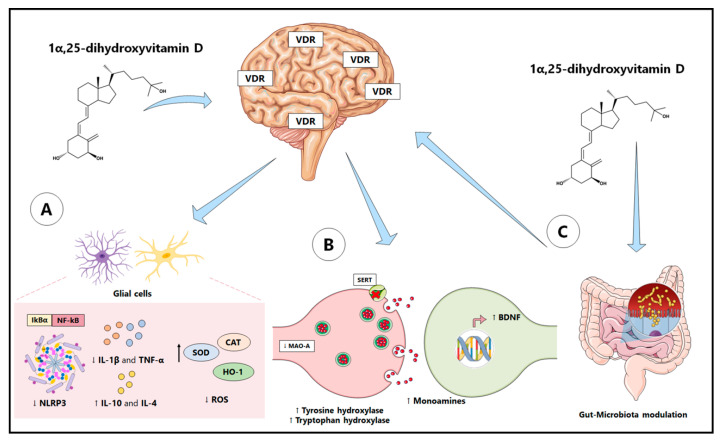
Antioxidant, anti-inflammatory, and neuromodulatory properties of vitamin D. Neurons and glial cells are able to express VDR in regions such as the prefrontal cortex and hippocampus. In these regions, the positive modulation of antioxidant enzymes, such as HO-1, CAT, and SOD, by calcitriol contributes to redox homeostasis and, consequently, to the attenuation of the neuroinflammatory process. Calcitriol is also able to increase the expression of IkBα, thus, inhibiting the nuclear translocation of NF-kB. As a consequence, less synthesis, oligomerization, and activation of the NLRP3 inflammasome occurs, resulting in attenuation of pro-inflammatory cytokines (such as IL-1β and TNF-α) and an increase in anti-inflammatory cytokines (such as IL-10 and IL-4) (**A**). In addition, vitamin D has also been shown to modulate the intestinal microbiota (**C**). Finally, vitamin D actively regulates the synthesis of monoamines and BDNF, thus, favoring the process of synaptic plasticity (**B**). Abbreviations: ↓: decreased; ↑: increased; BDNF: brain-derived neurotrophic factor; CAT: catalase; HO-1: heme oxygenase-1; IL-1β: interleukin-1β; IL-4: interleukin-4; IL-10: interleukin-10; IkBα: nuclear factor of kappa light polypeptide gene enhancer in B-cells inhibitor, alpha; MAO: monoamine oxidase; NF-kB: nuclear factor kappa B; NLRP3: NOD-like receptor family pyrin domain-containing 3; ROS: reactive oxygen species; SERT: serotonin transporters; SOD: superoxide dismutase; TNF-α: tumor necrosis factor-alpha; VDR: vitamin D receptor.

**Table 1 ijms-23-07077-t001:** Preclinical studies on effects of vitamin D in models of depression and anxiety.

Study	Animal Model	Treatment	Behavioral Alterations	Biochemical Alterations
Fedotova et al., 2016 [92]	Ovariectomized Wistar rats	Cholecalciferol (5 mg/kg for 14 days; s.c.)	Antidepressant-like effect in the FST	Not evaluated
Fedotova et al., 2017 [10]	Ovariectomized Wistar rats	Cholecalciferol (5 mg/kg for 14 days; s.c.)	Anxiolytic-like effect in EPM and LDT	Not evaluated
Fedotova 2019 [93]	Ovariectomized Wistar rats	Cholecalciferol (5 mg/kg for 14 days; s.c.)	Anxiolytic-like effect in EPM and LDT	Not evaluated
Camargo et al., 2018 [94]	Corticosterone (21 days) in male Swiss mice	Cholecalciferol (2.5 µg/kg, for 7 days; p.o.)	Antidepressant-like effect in the splash test and TST	↓ Protein carbonyl and nitrite levels
da Silva Souza et al., 2020 [95]	Corticosterone (21days) in female Swiss mice	Cholecalciferol (100 IU/kg, p.o.) for 7 days	Antidepressant-like effect in the TST	↓ ROS
Camargo et al., 2020 [57]	Corticosterone (21 days) in male Swiss mice	Cholecalciferol (2.5 μg/kg, p.o.) for 7 days	Antidepressant-like effect in the TST and splash test	↓ ASC, caspase-1, and TXNIP
Neis et al., 2022 [96]	CUMS in female Swiss mice	Cholecalciferol (2.5 μg/kg, p.o.) for 7 days	Antidepressant-like effect in the TST	↑ serotonin levels in the prefrontal cortex
Zhang et al., 2020 [97]	Ovariectomized female Sprague-Dawley rats	Calcitriol (100 ng/kg, p.o.) for 10 weeks	Antidepressant-like effect in the FST and novelty-suppressed feeding test	↓ IL-1β, IL-6, and TNF-α, iNOS and COX-2
Bakhtiari-Dovvombaygi et al., 2021 [98]	CUMS in male Wistar rats	Vitamin D_3_ (10,000 IU/kg) for 28 days	Anxiolytic and antidepressant-like effect in EPM and FST	↓ Malondialdehyde and IL-6 ↑ Total thiols, SOD, and CAT
Jiang et al., 2013 [99]	CUMS (4 weeks) in male Sprague-Dawley rats	Without treatment	Depressive-like behavior in sucrose preference test	↑ 1,25(OH)_2_D and VDR
Koshkina et al., 2019 [76]	CUMS in ovariectomized Wistar rat	Vitamin D_3_ (5 mg/kg for 4 weeks; s.c.)	Antidepressant-like effect in FST and sucrose preference test	↑BDNF and NT-3/NT-4
Xu and Liang 2021 [75]	post-stroke depression model (male C57BL/6 mice)	Calcitriol (25 μg/kg/day for 4 weeks; i.c.v.)	Antidepressant-like effect in FST and sucrose preference test	↑ VDR and BDNF

↓ decreased; ↑ increased; ASC: apoptosis-associated speck-like protein containing a caspase recruitment domain (CARD); BDNF: brain-derived neurotrophic factor; CAT: catalase; COX-2: cyclooxygenase-2; CUMS: chronic unpredictable mild stress; EPM: elevated plus maze; FST: forced swimming test; i.c.v.: intracerebroventricular; IL-1β: interleukin-1β; IL-6: interleukin-6; iNOS: inducible nitric oxide synthase; LTD: light–dark box test; p.o.: by mouth (orally); ROS, reactive oxygen species; s.c.: subcutaneous; SOD: superoxide dismutase; TNF-α: tumor necrosis factor-alpha; TST: tail suspension test; TXNIP: thioredoxin-interacting protein; VDR: vitamin D receptor.

## Data Availability

Not applicable.

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
