# Peer review of "Molecular Basis Underlying the Therapeutic Potential of Vitamin D for the Treatment of Depression and Anxiety"

_ijms, 2022, doi:10.3390/ijms23137077_

Round 1

Reviewer 1 Report

This is a well-written, well-thought, and sound review about the potential of vitamin D for the treatment of depression and anxiety. After a brief introduction to the problem of depressive and anxiety disorders, the Authors discussed the main mechanisms that may underlie the potential antidepressant and anxiolytic effects of vitamin D. They also referred to original pre- and clinical studies that support the therapeutic potential of the compound for the management of these disorders. I consider the strengths of the work to be: an interesting topic, a rich range of literature (115 references published up to June 2022), and good readability. In my opinion, the manuscript can be interesting for the readers of IJMS and I recommend it for publication in its present form.

Author Response

This is a well-written, well-thought, and sound review about the potential of vitamin D for the treatment of depression and anxiety. After a brief introduction to the problem of depressive and anxiety disorders, the Authors discussed the main mechanisms that may underlie the potential antidepressant and anxiolytic effects of vitamin D. They also referred to original pre- and clinical studies that support the therapeutic potential of the compound for the management of these disorders. I consider the strengths of the work to be: an interesting topic, a rich range of literature (115 references published up to June 2022), and good readability. In my opinion, the manuscript can be interesting for the readers of IJMS and I recommend it for publication in its present form.

Response: We thank the Reviewer for their positive feedback on our review manuscript. We are very pleased that the Reviewer found it worth of publication and a valuable addition to the field.

Reviewer 2 Report

This work is very nicely written and comprises a comprehensive review of the topic area reflected in the heading; MOLECULAR BASIS UNDERLYING THE THERAPEUTIC POTENTIAL OF VITAMIN D FOR THE TREATMENT OF DEPRESSION AND ANXIETY.

Vitamin D is an inexpensive and easily modifiable factor that has potentially many benefits, such as in the treatment of depression and anxiety, although so far findings from clinical trials are inconsistent. The review provides convincing support to justify the further pursuit of a potential role for vitamin D in the treatment of anxiety and depression, based upon preclinical trials and molecular mechanisms, with some limited support from clinical studies.  The article provides a useful and comprehensive resource and contribution to the field. It is noted that the senior author published a review in this area in 2019, however the focus and content is sufficiently diverse, with the current manuscript focussing more on molecular mechanisms.

Minor comments:

In the Literature Data Searching section, it should be mentioned up until what date the articles for review were collected.

This review provides alot of (convincing) information regarding molecular mechanisms of therapeutic benefit, however clinical trials have not consistently shown beneficial effects and in fact some studies have suggested the existence of reverse causality, at least for depression. In the section on clinical trials, after describing the literature, the authors could consider mentioning some of the potential reasons why clinical trials have not shown a consistent benefit in light of the molecular mechanisms described in the review, and if possible, recommend the best way forward from here. 

At line 152- the use of 'Therefore' appears inappropriate.

At line 230, typo in Bacteroidetes

At line 327, improved should be 'improve'.

Author Response

This work is very nicely written and comprises a comprehensive review of the topic area reflected in the heading; MOLECULAR BASIS UNDERLYING THE THERAPEUTIC POTENTIAL OF VITAMIN D FOR THE TREATMENT OF DEPRESSION AND ANXIETY.

Vitamin D is an inexpensive and easily modifiable factor that has potentially many benefits, such as in the treatment of depression and anxiety, although so far findings from clinical trials are inconsistent. The review provides convincing support to justify the further pursuit of a potential role for vitamin D in the treatment of anxiety and depression, based upon preclinical trials and molecular mechanisms, with some limited support from clinical studies.  The article provides a useful and comprehensive resource and contribution to the field. It is noted that the senior author published a review in this area in 2019, however the focus and content is sufficiently diverse, with the current manuscript focussing more on molecular mechanisms.

Response: We thank the Reviewer for their positive feedback on our review manuscript. We are very pleased that the Reviewer found it worth of publication and a valuable addition to the field.

Minor Comments:

  1. In the Literature Data Searching section, it should be mentioned up until what date the articles for review were collected.

Response: We thank the reviewer for the suggestion. Please note that the date of the articles used in this review has been added to Section 2 (Literature Data Searching). The new sentence reads as follows:

To review the molecular basis underlying the therapeutic potential of vitamin D for the treatment of depression and anxiety, we selected preclinical and clinical studies published over a 28-year period (1994 to 2022).”

  1. This review provides a lot of (convincing) information regarding molecular mechanisms of therapeutic benefit, however clinical trials have not consistently shown beneficial effects and in fact some studies have suggested the existence of reverse causality, at least for depression. In the section on clinical trials, after describing the literature, the authors could consider mentioning some of the potential reasons why clinical trials have not shown a consistent benefit in light of the molecular mechanisms described in the review, and if possible, recommend the best way forward from here. 

Response: We thank the reviewer for raising this interesting issue. Please note that we have now added a paragraph at the end of Section 6 (Clinical Studies: Effects of Vitamin D in Depression and Anxiety) to address the possible factors that may contribute to the discrepant results that have been reported in clinical trials. This new paragraph reads as follows:

Overall, although there is compelling clinical evidence pointing to the benefits of vitamin D for the management of depression and anxiety, it is important to note that divergent results have also been obtained [47]. Multiple factors may contribute to these discrepant results, including differences in the doses of vitamin D used, treatment time, serum 25-hydroxyvitamin D levels at baseline, nutritional condition at the onset of the treatment, age and sex of the individuals, as well as the presence of comorbidities that may influence the efficacy of vitamin D supplementation. Additionally, the heterogeneity observed in clinical studies may also be associated with genetic polymorphisms that may affect vitamin D efficacy [116-118]”.

In addition, the following sentence has also been included in Section 7 (Conclusions and Future Directions): “The discrepant results from clinical trials call for the need to conduct future studies so as to establish a protocol of vitamin D supplementation that is effective in preveningt or attenuating depressive and anxiety symptoms”.

  1. At line 152- the use of 'Therefore' appears inappropriate.

Response: We thank the Reviewer for pointing out this grammatical error, which has now been corrected.

  1. At line 230, typo in Bacteroidetes.

Response: Thank you. Please note that this typo has been corrected.

  1. At line 327, improved should be 'improve'.

Response: Thank you. Please note that this typo has been corrected.
